# An Ultrasensitive and Selective Determination of Cadmium Ions at ppt Level Using an Enzymic Membrane with Colorimetric and Electrochemical Detection

**DOI:** 10.3390/bios12050310

**Published:** 2022-05-07

**Authors:** Raouia Attaallah, Aziz Amine

**Affiliations:** Laboratory of Process Engineering and Environment, Faculty of Sciences and Techniques, Hassan II University of Casablanca, Mohammedia 21100, Morocco; raouia.attaallah-etu@etu.univh2c.ma

**Keywords:** enzymic membrane, cadmium ion, medium exchange, irreversible inhibition, reversible inhibition, electrochemical biosensor, drinking water

## Abstract

Cadmium ions (Cd^2+^) are extremely toxic heavy metal pollutants found in the environment, and which endanger human health. Therefore, it is critical to develop a sensitive and simple method for rapidly detecting Cd^2+^ in water samples. Herein, an enzymic membrane was developed based on an easy and rapid immobilization method of horseradish peroxidase (HRP), for determination of Cd^2+^ in drinking water. Hence, for the first time, an enzymic membrane was applied for the detection of Cd^2+^ without being pretreated. In the first format, the inhibition of horseradish peroxidase was performed using a colorimetric microplate reader. Under optimal conditions, the achieved limit of detection was 20 ppt. In addition, an electrochemical biosensor was developed, by combining the enzymic membrane with screen printed electrodes, which showed a linear calibration range between 0.02–100 ppb (R^2^ = 0.990) and a detection limit of 50 ppt. The use of this enzymic membrane proved to be advantageous when reversible inhibitors such as the copper ion (Cu^2+^) were present in water samples, as Cu^2+^ can interfere with Cd^2+^ and cause erroneous results. In order to alleviate this problem, a medium exchange procedure was used to eliminate Cu^2+^, by washing and leaving only cadmium ions as an irreversible inhibitor for identification. The use of this membrane proved to be a simple and rapid method of immobilizing HRP with a covalent bond.

## 1. Introduction

Excessive levels of heavy metals are a serious threat to human health and the ecosystem. Heavy metals are increasing dramatically, due to rapid industrialization and innovative farming practices [1]. Various industrial sources, including pesticide industries, electroplating industries, tanning industries, and chemical industries, have contributed to the deterioration of water quality with high levels of heavy metal ions [2]. Cadmium ion (Cd^2+^) is among the most hazardous heavy contaminants in water, and it is one of the most dangerous to human health [3,4,5]. Excess Cd^2+^ consumption can result in cancer, cardiovascular disease, and liver and kidney damage [6,7]. The solubility of Cd^2+^ is much higher than other heavy metals, and it can be rapidly transferred to bio-systems [8]. In this respect, the United States Environmental Protection Agency (EPA) and the World Health Organization (WHO) established 0.005 mg/L and 0.003 mg/L as the maximum contaminant level (MCL) of Cd^2+^in drinking water, respectively [9,10].

Nowadays, the detection of Cd^2+^ in drinking water samples represents an important topic [11]. There are many classical analytical techniques employed for qualitative and quantitative detection of cadmium ions in environmental and biological samples, including atomic fluorescence spectroscopy (AFS) [12], atomic absorption spectroscopy (AAS) [13], and inductively coupled plasma mass spectrometry (ICP-MS) [14]. The above-mentioned techniques have various positive features, such as a high selectivity, sensitivity, and accuracy. However, they present some shortcomings, such as their complexity, cost, and time-intensive steps of pre-concentration and analysis; more importantly, these methods cannot be applied in the field and for real-time testing. For these reasons, establishing a rapid, selective, and sensitive method for Cd^2+^ is of great importance.

Enzymes are widely integrated into biosensors and bioassays for sensitive determination of substrates, inhibitors, and/or activators [15,16,17]. Among the various enzymes, HRP inhibition has been widely studied in the field of colorimetric and electrochemical methods for the screening of a wide range of inhibitors, including cadmium ions. [18,19]. Furthermore, enzyme inhibition is a promising approach for detecting the bioavailable fraction of Cd^2+^ in water samples in a simple and rapid manner. Lately, membranes have received much interest as potential materials, because they are simple, inexpensive, user-friendly, and require a minimal sample, and also are used as a carrier for enzyme immobilization, as well as selective membranes [20,21,22]. Membrane modification techniques aim to introduce functional groups onto the membrane surface, thereby improving the membrane’s surface properties. Some of these functional groups allow enzymes to be immobilized on membranes [23]. In addition, the most common detection method used with enzymes immobilized in the membrane (enzymic membrane) is colorimetry, where specific reagents are immobilized on the membrane, and the developed color intensity is correlated with analyte concentration [24]. In contrast, colorimetric methods are convenient and attractive for many applications, because they can be easily monitored by the naked eye, without the requirement for advanced and complicated instruments [25,26]. Electrochemical detection has also been used with enzymic membranes and generates more quantitative results [27]. Previously, we reported a colorimetric detection of Cd^2+^ based on the irreversible inhibition of HRP immobilized on filter paper, where a detection limit of 0.1 ppb was obtained [18]. Currently, the greatest challenge in developing a cadmium ion selective method is to overcome the interference of other transition metal ions, especially copper ions (Cu^2+^), due to their inhibition of HRP. Thus, the development of a platform that can effectively and selectively recognize Cd^2+^, without interference from Cu^2+^ remains a challenging field of research.

In the present work, a pre-activated immunodyne membrane was used as a solid substrate to immobilize horseradish peroxidase for the detection of Cd^2+^ in water samples, while avoiding interference with Cu^2+^. To the best of our knowledge, this is the first time an immunodyne membrane based on the immobilization of an enzyme has been developed to detect traces of heavy metals. In the first format, a bioassay based on an enzymic membrane was developed for ultrasensitive and selective detection of cadmium ions. Furthermore, the development of a biosensor by immobilizing horseradish peroxidase within the porous structure of a membrane placed on the top of SPEs, as a sensitive and simple device, was successfully performed. As confirmed by cyclic voltammetry (CV), horseradish peroxidase was inhibited by small quantities of cadmium ions, using TMB (3,3′,5,5′-Tetramethylbenzidine) as an electroactive substrate. In both cases, the developed platform was successfully employed to detect low cadmium ion concentrations. 

## 2. Materials and Methods

### 2.1. Chemicals and Reagents

All chemicals used in this work were of analytical grade and used without further purification. Sodium acetate buffer solution (0.1 M, pH 5) was prepared from sodium acetate and acetic acid. Phosphate buffer solution (0.1 M, pH 7) was prepared from sodium hydrogen phosphate and sodium dihydrogen phosphate. 3,3′,5,5′-Tetramethylbenzidine (TMB) ready to use supersensitive liquid chromogenic substrate, containing 1.25 mM TMB and 2.21 mM H_2_O_2_, was purchased from Sigma-Merck, Germany. Horseradish peroxidase (EC 1.11.1.7 type VI-A from horseradish 1280 units mg^−1^, type XII from horseradish 1000 units mg^−1^), trehalose, and EDTA were also purchased form Sigma-Merck, Darmstadt, Germany.

The standard solution of Cd^2+^ was purchased as an atomic absorption standard solution (1000 µg/mL) from Alfa Aesar, Kandel, Germany, and Cu^2+^ was purchased as an atomic absorption standard solution (1000 mg/L) from Sigma-Aldrich, Darmstadt, Germany.

The pre-activated immunodyne ABC nylon membrane (⌀ = 3 µm) was supplied by Pall Europe Limited, Portsmouth, UK. 

Screen-printed electrodes (SPEs) were obtained as a gift from professor F. Arduini of Tor-Vergata University (Italy), they consisted of a graphite working electrode, a silver/ silver chloride reference electrode (Ag/AgCl), and a counter electrode; 3 mm in diameter and with an active surface of about 0.07 cm^2^. 

### 2.2. Apparatus

An automated ELx800 absorbance microplate reader (BioTek Instruments, Winooski, VT, USA) was used to measure the optical density of the developed colorimetric product at 450 nm. Data were evaluated with Gen5 software. Electrochemical measurements were carried out with a PalmSens 4 potentiostat (PalmSens BV, Houten, The Netherlands) controlled by a PC running PSTrace 5.0 software. OriginPro8 was used as software for data analysis, processing, and graphing.

### 2.3. Preparation of the Enzymic Membrane

The immunodyne ABC membrane is a nylon membrane designed for in vitro diagnostic applications, whose surface is modified with reactive groups to covalently bind enzymes. It is intrinsically hydrophilic and has a high density of covalent binding sites, which are able to react with the OH, COOH, or NH_2_ groups present in biomolecules [28].

The immobilization of horseradish peroxidase (HRP) was performed by immersing a small piece of the pre-activated membrane (0.8 cm) in phosphate buffer (0.1 M, pH 7) containing HRP for 30 min at 4 °C, to allow coupling between the enzyme and the membrane. Afterwards, the enzyme-loaded membrane was washed several times using phosphate buffer to remove the unbound enzyme. The enzymic membrane was then allowed to air dry before being used.

### 2.4. Enzymatic Assays

#### 2.4.1. Spectrophotometric Assay 

The inhibition of horseradish peroxidase was evaluated in the presence of different Cd^2+^ concentrations, and by measuring the absorbance values of the enzymatic product at 450 nm using a microplate photometer. In detail, as illustrated in Appendix A, the enzymic membrane was placed into a microplate well, then, 50 µL of acetate buffer was added. Prior to adding the chromogen substrate, HRP enzyme was incubated with 30 µL of various Cd^2+^concentrations for 20 min at room temperature. Afterward, 20 µL of TMB was added to the reaction. The mixture was then incubated at room temperature for 2 min with a total volume of 100 µL. The enzymatic reaction was stopped with 50 µL of 1M H_2_SO_4_, and the enzymic membrane was removed from the microplate wells. Finally, the absorbance of the yellow color developed upon the reaction was measured at 450 nm. 

#### 2.4.2. Electrochemical Horseradish Peroxidase Assay 

Prior to the modification, screen printed electrodes (SPEs) were pre-treated in PBS solution (0.05 M, pH 7) with 0.1 M KCl, using an anode polarization potential of +1.7 V for 3 min. It was reported by Ricci et al. [29] that electrochemical pre-treatments of SPEs can significantly enhance their electrochemical performance. A bioactive disc of a size corresponding to the electrodes was cut from the enzymic membrane and stored at 4 °C in phosphate buffer until use. The enzyme loaded membrane was placed on top of the SPEs, which was held horizontally, to completely cover the working, counting, and reference electrodes. After incubating different cadmium ion concentrations with the enzymic membrane, 20 µL of ready-to-use TMB was added to the reaction. Then, the mixture was incubated at RT for 5 min. Once the reagents have been introduced to the enzymic membrane, it attaches to the electrode without the use of tape, due to its hydrophilicity. All experiments were conducted at RT (22 °C).

### 2.5. Electrochemical Measurements

Cyclic voltammetry (CV) was used to measure the electrochemical behavior of the developed biosensor in 0.1 M acetate buffer solution pH 5 of 0.1 M KCl. The measurements were carried out at a scan rate of 50 mV·s^−1^ and a potential window of −0.2 V to +0.6 V.

### 2.6. Reversible and Irreversible Inhibition of Horseradish Peroxidase 

The mechanism of enzymatic inhibition can be irreversible or reversible (Appendix A). The inhibitor is attached by strong bonds or firmly entrapped to the active site of the enzyme in irreversible manner, resulting in the permanent inactivation of the enzyme. As a result, the enzyme’s original activity is lost and cannot be recovered. In general, irreversible inhibition necessitates a long incubation period and a low concentration of enzyme. On the other hand, a reversible inhibitor can inactivate an enzyme via a weak bond and freely dissociate from the enzyme; bearing in mind that in the case of an immobilized enzyme, simple washing steps can easily distinguish between reversible and irreversible inhibition. However, in the case of irreversible inhibition the enzyme cannot regenerate its initial activity, unless some specific reagents are added. To evaluate the degree of inhibition (*I*%) of HRP activity, the electrochemical signal or absorbance responses in the absence (*I*_0_) and the presence of an inhibitor (*I*_1_) were measured, with either an electrochemical or colorimetric method. The degree of inhibition was expressed as follows:(1)$$I\left(\%\right)=\frac{I_{0}-I_{1}}{I_{0}}\times 100$$
where *I*_0_ and *I*_1_ are the current or absorbance recorded before and after inhibition, respectively. 

### 2.7. Medium Exchange Procedure

The presence of various ions and compounds in the matrix sample can interfere with cadmium ion detection, resulting in inaccurate results. The application of the medium exchange procedure can reduce drastically the interference and greatly increase the selectivity, by discriminating irreversible inhibitors (cadmium ions) from reversible inhibitors (such as copper ions). The developed enzymic membrane was first incubated in a matrix sample containing both Cd^2+^ and Cu^2+^. Then, the enzymic membrane was thoroughly washed to remove Cu^2+^and leave only Cd^2+^, which was attached to the enzyme by strong bonds and/or firmly entrapped in the enzyme’s active site. Finally, the enzymic membrane was immersed in acetate buffer (medium exchange) for cadmium monitoring (Appendix A).

## 3. Results and Discussion

### 3.1. Spectrophotometric Assay

In the present section, a colorimetric enzymic membrane was developed for the sensitive detection of cadmium ions in contaminated drinking water. Horseradish peroxidase (HRP) could oxidize 3,3′,5,5′-tetramethylbenzidine (TMB) substrate and induce a blue colored solution, followed by addition of sulfuric acid to stop the reaction, which lead to a yellow color solution, corresponding to an absorbance at 450 nm. To immobilize the enzyme on the membrane, different immobilization times (10, 30, and 60 min were tested to define the requested time for the immobilization of the enzyme on the pre-activated membrane. As stated in Appendix A, the absorbance values increased from 0.650 to 0.880 and 0.900 when increasing of immobilization time from 10 to 30 and 60 min, respectively, and thus, the absorbance values barely changed between 30 and 60 min. To avoid a lengthy incubation period, 30 min was chosen as the appropriate time for the remaining experiments, since there was no notable change between 30 and 60 min.

The addition of the inhibitor caused it to bind to the active site of the enzyme, leading to low color development and, thus, low absorbance values, measured by a microplate reader. A schematic illustration of the spectrophotometric assay is depicted in Appendix A. 

The experimental conditions that can affect the performance of the inhibition of HRP, such as the substrate and enzyme concentration, pH of supporting electrolyte, reaction time, and incubation time were studied, in order to optimize the inhibition response to cadmium ions. According to previous works, the optimal pH for the enzymatic reaction of HRP is pH 5 in acetate buffer solution [18,19], in agreement with the pH recommended for the use of TMB chromogenic substrate by the company Sigma-Merck, Darmstadt, Germany. [30]. Therefore, pH 5 was chosen for further experiments. Since the inhibition of HRP by cadmium ions is of irreversible type, the concentration of HRP is one of the most important parameters that directly affect the degree of inhibition [31]. Therefore, the HRP concentration was optimized by testing three different concentrations, namely, 0.1, 0.2, and 1 µg/mL; to achieve high sensitivity, the enzyme should be chosen as the minimum amount of enzyme capable of producing a measurable signal and the lowest amount of enzyme necessary to achieve the lowest detection limit. As shown in Figure 1A, a high degree of inhibition was obtained with 0.1 µg/mL of HRP. Thus, 0.1 µg/mL was selected for the remainder the work. Moreover, different concentrations of the TMB/H_2_O_2_ substrate were tested. As depicted in Figure 1B, to achieve a better sensitivity and selectivity, and, on the other hand, to avoid extremely high absorbance values, 0.625 mM TMB and 1.105 mM H_2_O_2_ were selected as the optimal concentrations of TMB/H_2_O_2_ for the subsequent experiments. The next parameter optimized was the reaction time, which is the time of reaction between the enzyme and the substrate needed to obtain the enzymatic product. The effect of three different reaction times was investigated (1, 2, and 4 min). As demonstrated in Figure 1C, the reaction time selected was 2 min, as the best compromise regarding the sensitivity and time of analysis. The effect of incubation time on the degree of inhibition was studied through incubating the HRP with Cd^2+^ and measuring the inhibition over time. In fact, in the case of irreversible inhibition, the degree of inhibition increased with an increased time of incubation [31,32]. As depicted in Figure 1D, various incubation times were tested (10, 20, 30, and 40 min). For irreversible inhibition, it is possible to achieve lower detection limits using longer incubation times; in fact, the degree of enzyme inhibition increases with the incubation time, until reaching a plateau. The results showed that the highest degree of inhibition was obtained with 40 min. This also confirmed that the inhibition of HRP by cadmium ions was irreversible. As a compromise between the degree of inhibition and time of analysis, 20 min was selected as the time of incubation for the rest of the work.

Three types of enzyme were tested under the same experimental conditions, to determine the type that was most sensitive to cadmium ions (horseradish peroxidase type VI-A, type XII, and antibody labelled horseradish peroxidase). Indeed, each of these enzymes was immobilized on the membrane at the proper concentration and then incubated with 10 ppb and 100 ppb of cadmium ions under the same conditions. All three types of enzyme are inhibited by cadmium ions, as indicated in Table 1. Although these three enzymes are from horseradish roots, they are purified and conditioned following different protocols, which may affect their sensitivity toward cadmium inhibitor. As a result, HRP type VI-A was the most sensitive, being inhibited by more than 50% with only 10 ppb of cadmium ions. This demonstrates the rationale for choosing this enzyme for this work.

The inhibiting potency of Cd^2+^ on the oxidation of 3,3′,5,5′-tetramethylbenzidine (TMB) by HRP immobilized on the membrane was determined by measuring the intensity of the yellow-colored solution (after stopping the reaction with H_2_SO_4_) at 450 nm using a microplate reader. The degree of inhibition was found to vary with the concentration of cadmium ions. The calibration curve in Figure 2 shows a high slope at a low concentration of Cd^2+^ and becomes flat at a concentration higher than 20 ppb. Thus, the degree of inhibition was 16%, 25%, 32%, and 61 %, at 0.02, 0.1, 0.5, and 100 ppb of Cd^2+^, respectively. The inset of Figure 2 illustrates the good linearity of the low level of cadmium ion concentrations in the range 0.02–100 ppb. The regression equation is I=36.51+12.24log(c ), in which *I* is the degree of inhibition (%) and *c* is the concentration of Cd^2+^ (ppb), R^2^ = 0.998. The achieved detection limit under the experimental conditions was 0.02 ppb, based on a IC_10_ value, which is a 10% inhibitory concentration. It was proposed that IC_10_ can be considered as a practical LOD of the degree of inhibition [31]. The obtained detection limit is much lower than the maximum concentrations established by the EPA (5 ppb) and WHO (3 ppb) [9,10]. The results demonstrated that the developed enzymic membrane could accurately discriminate Cd^2+^ concentration, which makes it suitable for monitoring low contents of Cd^2+^ in water simples. The reproducibility was good, since similar results (RSD = 2%) were achieved for the five replicate analyses. 

### 3.2. Electrochemical Biosensor

The developed enzymic membrane was placed on top of the SPEs, to completely cover the electrodes surface, as depicted in Figure 1, and, thus, to construct the developed electrochemical biosensor. Furthermore, the developed biosensor was analyzed by cyclic voltammetry (CV), using the electroactive TMB substrate, because it is less toxic and provides higher assay sensitivity and faster reactions than other HRP substrates, such as *O*-phenylenediamine (OPD) or 2,2′-azino-bis-(3-ethylbenzthiazoline-6-sulfonic acid (ABTS) [33], and it offers the possibility of performing either optical or electrochemical measurements. In this work, we used a commercial preparation of TMB that is commonly used in ELISA assays; this preparation contains hydrogen peroxide (H_2_O_2_) and a series of unknown additives (the supplier would not disclose the composition of the mixture). We chose to use this commercial mixture because the proportions of TMB and H_2_O_2_ and solvent composition in it have been optimized to provide better signals in the presence of HRP. Cyclic voltammetry was chosen because the entire peaks were recorded at the same scan and the measurement of the value of one of these peaks was sufficient to evaluate the inhibition of horseradish peroxidase by cadmium at ppt level.

The electrochemical behavior of SPEs/HRP in 0.1 M acetate buffer solution (pH 5) containing 0.1 M KCl was investigated using cyclic voltammetry. The TMB underwent a two-electron oxidation-reduction process [34], which was confirmed by cyclic voltammetry using our biosensor. Figure 3 shows the change of redox peak current, which indicates the reaction between HRP and the substrate, and then HRP with cadmium ion and the substrate. The TMB substrate exhibited two oxidation peaks at +0.256 V and +0.432 V (Vs. Ag/AgCl), and two reduction peaks at +0.354 V and +0.214 V. When HRP was involved in the redox reaction of TMB, the CV continued to show two oxidation and two reduction peaks. However, compared to the CV observed when HRP was not involved in the reaction, the peak potential remained unchanged and both oxidation and reduction peaks decreased; suggesting that the oxidized TMB substrate (TMB_ox_) was completely reduced to a non-electroactive product within the range of potentials investigated.

When the biosensor was incubated with 10 ppb of cadmium ions, both oxidation and reduction peaks appeared, and a significant increment of the current in comparison with the cyclic voltammetry of HRP was observed. These findings revealed that the additive Cd^2+^ concentration inhibited part of the enzymes, allowing the TMB to be oxidized in solution by the enzyme molecules that were not occupied by the inhibitor.

To improve the analytical performance of the biosensor, the experimental parameters were carefully optimized using the developed biosensor, as summarized in Table 2. Different concentrations of TMB/H_2_O_2_ substrate were tested. However, in the presence of a high substrate concentration, it was difficult to distinguish between the TMB redox peak and that of the enzymatic reaction. Thus, the low substrate concentration 0.25/0.442 was selected as the optimal amount of TMB/H_2_O_2_ for the subsequent experiments. Afterwards, the HRP concentration was optimized by testing four different concentrations, namely 0.05, 0.1, 0.5, and 1 µg/mL, to achieve a high degree of inhibition, and producing a measurable signal with high precision. As shown in Table 2, a high degree of inhibition was obtained with 0.05 µg/mL of HRP. However, concentrations lower than this were not investigated because the sensitivity becomes very low and, thus, the precision is low as well. The next parameter optimized was the reaction time. Indeed, the effect of three different reaction times was investigated (1, 2, and 5 min). As indicated in Table 2, the sensitivity increased with the time of reaction. The reaction time selected was 5 min, as the best compromise regarding sensitivity and time of analysis. Finally, various incubation times were tested (10, 20, 30, and 60 min). In the case of irreversible inhibition, lower detection limits could be achieved using longer incubation times, as the degree of enzyme inhibition increased with incubation time. The results showed that the highest degree of inhibition was obtained with 60 min. As a compromise between the degree of inhibition and the analysis time, an incubation time of 20 min was chosen for further experiments. 

Once all parameters affecting the inhibition of horseradish peroxidase were optimized, the performance of the developed biosensor for the detection of Cd^2+^ using an enzymic membrane was investigated via cyclic voltammetry, which is the most widely used technique for electrochemical biosensor studies. Figure 4A shows cyclic voltammograms of the developed biosensor based on an enzymic membrane in different concentrations of Cd^2+^ (0.02–100 ppb). As expected, in the absence of the inhibitor, the current of the enzymatic reaction (HRP and TMB/H_2_O_2_) decreased. However, increasing the concentration of cadmium ions led to an increase in redox peak current intensity, caused by the reduction of TMB/H_2_O_2_.

The degree of inhibition was calculated as follows:(2)$$I\;\left(\%\right)=\left[\frac{\left(I_{TMB}-I_{0}\right)-\left(I_{TMB}-I_{1}\right)}{\left(I_{TMB}-I_{0}\right)}\right]\times 100$$
where *I_TMB_* is the signal of the substrate in the absence of the enzymatic reaction, *I*_0_ and *I*_1_ are the currents recorded before and after inhibition, respectively.

The resulting calibration curve is reported in Figure 4B. Thus, the degree of inhibition was 6%, 22%, 69%, and 77 %, at 0.02, 0.1, 50, and 100 ppb of Cd^2+^, respectively. The calibration curve covered the inhibitor range from 0.02 to 100 ppb, the regression equation is I=40.52+18.6log(c), in which *I* is the degree of inhibition (%) and *c* is the concentration of Cd^2+^ (ppb), R^2^ = 0.990 with high reproducibility (RSD = 3%). The achieved detection limit under the experimental conditions was 50 ppt based on the IC_10_ value, which is a 10% inhibitory concentration. This is significantly lower than the concentrations set by the EPA and WHO. The developed biosensor is extremely affordable and was constructed using a very simple fabrication technique.

The analytical performance of the developed bioassay (colorimetric/electrochemical) was compared to previously reported bioassays for the detection of cadmium ions. As shown in Table 3, the LOD for both of the present methods was lower than the previously reported works in the literature.

### 3.3. Selectivity Study

Copper ions have been identified as a major interference in the detection of Cd^2+^ in water. Therefore, the development of a simple approach for alleviating this interference challenge is of paramount interest. According to the published literature [39,41], copper ions are a reversible inhibitor of horseradish peroxidase. As copper ions can be weakly bound to the enzyme, they can be washed out, even after a long period of incubation with the enzyme membrane. (Figure 2A). The cadmium ion, on the other hand, is an irreversible inhibitor that becomes firmly entrapped in the active site and prevents any access of the substrate to the enzyme, and, thus, remains attached to the membrane, even after long and intense washing. (Figure 2B).

The selectivity study was performed with dual electrochemical and colorimetric detections in optimal conditions. For colorimetric detection, the enzymic membrane was first incubated with 10 ppb of Cd^2+^ and then washed for a few minutes with PBS buffer. As demonstrated in Appendix A, the obtained degree of inhibition was 41%; and after washing the enzymic membrane, the obtained degree of inhibition was the same. This confirms that cadmium is an irreversible inhibitor, as the active site is still occupied by cadmium ions after washing. However, as depicted in Appendix A, when the enzymic membrane was incubated with 50 ppm of copper ions, the obtained degree of inhibition was 22%. The membrane recovered 94% of its enzymatic activity after washing. This demonstrates that Cu^2+^ is a reversible inhibitor, because a simple wash removes almost all copper ions from the enzymic membrane.

It should be noted that various interfering species, with a concentration of 300 ppb, were tested, such as Pb^2+^, Ag^+^, Hg^2+^, and Zn^2+^, and which might be present in water samples with Cd^2+^. The findings revealed that these interfering species had no effect on the detection of cadmium ions using free HRP [18]. In addition, 125 mM NaCl was tested under our experimental conditions, and no inhibition effect was observed. The presence of 20 mM of EDTA in working buffer did not affect the degree of inhibition in the presence of Cd^2+^. This confirmed that the cadmium ion was firmly entrapped in the active site of the enzyme and prevented access of a substrate and chelating agent, such as EDTA. As a result, the restoration of enzyme activity was not possible, explaining the high potency of horseradish peroxidase inhibition by cadmium ions. 

The interference study was performed by introducing 50 ppm of Cu^2+^ into a standard solution containing 10 ppb cadmium ions. According to the findings (Figure 5), the degree of inhibition was 45%. As the copper ions were removed with washing, only cadmium ions could be measured after washing the enzymic membrane, yielding a degree of inhibition of 41 % related to cadmium ions. 

For the electrochemical detection under optimal conditions, the enzymatic membrane was incubated with 50 ppm of Cu^2+^, showing 49% inhibition. The enzymic membrane was able to regenerate its enzymatic activity before being washed with PBS buffer (Appendix A), demonstrating that copper ions are a reversible inhibitor of HRP. On the other hand, the enzymatic membrane was then incubated with Cd^2+^ and Cu^2+^, before being washed with PBS buffer for a couple of minutes. Figure 5B clearly shows that in the presence of 10 ppb Cd^2+^ and 50 ppm Cu^2+^, both the oxidation and reduction peaks increased compared to the CV of the enzymatic reaction (HRP-TMB). When the enzymatic membrane was washed, Cu^2+^ was removed from the enzyme, and the response was solely due to Cd^2+^.

To verify the practical applicability of the enzymic membrane, a water sample was collected from the university and then spiked with 10 ppb of Cd^2+^. The results showed that the % recovery of the spiked cadmium ions, before and after, were 110% and 98%, respectively. This enzymic membrane could be applied to determine cadmium ions in complex aqueous solutions. Compared with other methods of Cd^2+^ detection, this enzymic membrane showed a lower cost and better performance.

### 3.4. Stability of Enzymic Membrane

The storage stability of the enzymic membrane was investigated at 4 °C. The membrane was first coated with 0.1 µg/mL of HRP and then post-coated with 2% of trehalose as a stabilizing agent, which allowed maintaining the long-term stability of the biomolecules, as demonstrated in our previous work [18]; then the enzymic membrane was stored dry at 4 °C. The response of the enzymic membrane as a function of storage time was shown in Figure 6. After 30 days, dried enzymic membranes stored sealed with trehalose exhibited a linear response over a period of 30 days; however, 31% of the activity was lost. Dry storage of the membrane without stabilizing agent resulted in a complete loss of bound enzyme activity. The immobilized HRP on membrane was extremely low cost and straightforward, as no addition of chemical reagents was required (only trehalose as stabilizing agent), and is suitable for on-site applications. Thus, the nylon matrix membrane proved to be an effective and stable microenvironment for horseradish peroxidase.

## 4. Conclusions

In the present study, an enzymic membrane was designed for colorimetric and electrochemical biosensing of Cd^2+^ in drinking water at ppt/ppb level. Immobilization of horseradish peroxidase on an immunodyne membrane is extremely advantageous, because complex immobilization methods involving the use of multiple chemicals can be avoided, as well as for ensuring that the original enzyme configuration is not compromised, allowing for the optimal activity. Thanks to a medium exchange technique, the developed enzymic membrane could precisely quantify Cd^2+^ in water sample without significant interference from a high Cu^2+^ concentration (50 ppm), which can reversibly inhibit HRP. Moreover, the biosensor based on the enzymic membrane was highly advantageous, because of its simplicity and the cost-effectiveness of its preparation procedure, as well as the facile control of enzyme loading and good shelf life. Finally, this approach has great potential as a new tool for field measurement of Cd^2+^, particularly in resource limited settings.

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
