# Peer review of "An Ultrasensitive and Selective Determination of Cadmium Ions at ppt Level Using an Enzymic Membrane with Colorimetric and Electrochemical Detection"

_biosensors, 2022, doi:10.3390/bios12050310_

Round 1

Reviewer 1 Report

The authors of this manuscript proposed an interesting approach for sensitive and selective detection of Cd(II) ions in tap water using an enzymic membrane. For this, a commercially available immunodyne ABC membrane was used to attach the HRP enzyme, and both colorimetric and electrochemical assays were demonstrated, highlighting the versatility of the proposed enzymic membrane toward point-of-care analysis. However, there are some issues that should be addressed before this will be suitable for publication. Therefore, a major revision is necessary in this situation.

Comments:

  1. In the topic 3.1 (line 184) the authors mentioned that the absorbance values “barely changed” when increasing the immobilization time. I suggest adding some values instead, like percentage of immobilization to be more precise and quantitative. They also mentioned in line 186 that there was no notable change between 30 and 90 min. However, there is no 90 min in the graphic for the reader to witness it.
  2. From Fig. 1A we can see that 0.1ug/mL was the minimum value used. And seems like they could use even a more diluted HRP sample and get the same result. The authors should add some points to that graphic like: 0.02, 0.05 and 0.5 ug/mL for instance, for better choose and optimization of this important parameter.
  3. The same optimization problem mentioned in the comment 2 occurred in Fig. 1D, and the authors should add at least 1 more point to that graphic, like 50 or 60 min. They chose all the optimization parameters with only 3 points graphics. Some of them are ok, but if you are choosing one of the limit points of your graphic, then you should add more points to it, to ensure you are optimizing this parameter properly, or at least explain why not adding more point to the graphic. I have noticed the same thing in Table 2 regarding the enzyme concentration and reaction time.
  4. When testing the 3 HRP enzymes for the colorimetric assay (Table 1), it was not clear if the authors used all the optimized parameters of Figure 1 for all three enzymes. This should be added to the text.
  5. In Fig. 4, which peak was used to obtain the current to calculate the degree of inhibition? First or second? Anodic or cathodic? Why? It is also important to add the direction of the scanning cycle in the CV curve.
  6. CV is quite useful for characterizing the system. However, for detection and quantification chronoamperometry seems more likely, since it allows to maintain the electrode polarized in a specific potential so TMB will be oxidized only by the HRP enzymatic reaction and reduced by the electrode. Besides, it would simplify the needed apparatus for electrochemical analysis. Otherwise, the authors may add to the text what would be the importance of performing the whole cycle on the CV for the detection.
  7. I missed the curve “before washing” in Figure 5. This way we could see the effect of copper ions presence on the measured redox current.
  8. To demonstrate the real versatility of the proposed enzymic membrane, it would be better to use the membrane loaded with the same HRP enzyme for both colorimetric and electrochemical measurements. In other words, the same membrane that can be used in devices coupled to any of those two detection systems. Is it possible within the proposed system?

Reviewer 2 Report

The technology proposed by the author is very simple, which can be used for detection only by immersing ABC membrane in HRP for incubation. The performance of both colorimetric and electrochemical sensing is excellent. I think this work can be accepted for publication after solving the following problems.

  1. Since incubation time varies so much between 10 and 30 minutes, why didn't the author try 20 or 25 minutes? A satisfactory solution may be achieved in 20 minutes. It's not a huge difference for your experiment, but in practice it's important.
  2. There seems to be some inconsistency in font size in the text.
  3. According to the reported papers, the author believes that pH 5 can be selected. I think the author should explain further why this environment is more suitable for sensing.
  4. The authors selected three enzymes and found the best one. I think the author needs to give reasons. The author spends a lot of time on parameter optimization, but only describes the results without theoretical explanation.
  5. What are the corresponding oxidation peaks and reduction peaks?
  6. Do common contaminants in water other than copper ions not affect the detection of cadmium ions?

Reviewer 3 Report

Review:

An ultrasensitive and selective determination of cadmium ions 2 at ppt level using an enzymic membrane with colorimetric and 3 electrochemical detection

The manuscript is a very interesting analytical approach.  I believe overall the work is very well done.  I believe it is publishable, however I have some specific comments on aspects of the work that I believe should be clarified.  I would not consider these suggestions to be major revisions.

TMB should be defined on line 81

Line 116. Can a more exact number be placed on the immersion of the membrane in the HRP solution?  A couple of minutes in not very descriptive. Did the authors explore the reproducibility of the enzyme immobilization? Is there anyway to use color formation or protein staining to estimate the amount of enzyme immobilized on the membrane?  Some comments on reproducibility of the immobilization would be useful.

Is this really a field-testing method?  The microplate reader is not likely easily transported to the field. Similarly does the cyclic voltammetry require nitrogen purging of the cell?  If not, than the electrochemical method could be used remotely. Since the motivation is at least partially the development of a field screening method, what would need to be done make this an actual field method?  Can this measurement be made in a field transportable (battery operated) spectrophotometer?

What volume of water sample was processed during the exposure of the membrane to Cd2+? I understand that that the membrane must be rinsed to remove copper.   Is the membrane rinsed after Cd2+ exposure for all analysis (even without Cu) or is the TBD just added directly to the sample well?  What is the total volume in the well at the time of measurement?  If the medium exchange is the normal routine, does Cd2+ possible cause the enzyme to detach from the membrane? Could that contribute to the loss of the color generation?

Can the authors describe the electrochemical cell? What is the cell volume?  Is the SPE in a horizonal position and is the membrane exposed to the Cd2+ while in the electrochemical cell?  I don’t understand the authors statement on the attachment of the membrane to the electrode?  What is “pence”?  Does the membrane simply self-adhere to the electrode?  The description is not very clear.

I don’t understand Figure 1B. What is the concentration of TMB. It appears that the authors are reporting a dilution of the solution purchased from Merck?  What is the original TMB concentration? Is this the same solution used for the electrochemical method?  It seems that the concentration could be estimated by using uv/vis on the original solution. If you can only report the dilution of the product that should be made more clear and the dilution should not be referred to as a concentration.

What Cd2+ concentration was used to generate Figure 1 D?

The authors indicate the impact of Cd2+ on the enzyme is irreversible.  They state that a covalent bond is formed between the enzyme and the HRP. Is it possible that Cd2+ just causes irreversible denaturation of the HRP, whereas Cu2+ does not?  It is hard to imagine a covalent Cd compound (organo Cd) forming in an aqueous environment. Is there specific evidence for a particular association of Cd2+.

I don’t understand the argument that slow binding kinetics implies the formation of a covalent bond.  Perhaps the changes in tertiary structure of the enzyme induced by Cd2+ are simply slow.

It appears that 100% inhibition is never achieved under these conditions.  This implies that the enzyme is perhaps becoming less efficient due to the interaction with Cd2+ but not completely non-catalytic.  Perhaps the available enzyme is becoming “saturated” with Cd2+ which leads to the observed plateau in figure 2?  Could the range of the assay be changed by attaching more enzyme to the membrane?

In figure 2 the response is not linear, but apparently log linear rather than linear as stated in the manuscript.  Were any other types of plots of the data attempted? Can this type of response be rationalized based on reaction mechanism?   

Keyhani et al., (Biochimica et Biophysica Acta 1621 (2003) 140– 148) has studied Cd2+ binding to HRP(at much high concentrations). They appear to conclude that Cd2+ induces a change in enzyme structure, thus impacting catalytic efficiency.  Although they reported changes in light absorption of the enzyme after Cd2+ exposure,  I don’t believe they saw any evidence for the loss of Iron from HRP heme group.

The authors are able to remove Cu2+ by washing.  What about the effect of other metals that might be present such as Pb, Zn or Hg etc.? Since they are not measuring metal concentration in the protein there is no way to know if the Cd2+ was removed as well.  Even if the Cd2+ is removed it may be that conformational changes induced are not reversible.   I believe without further evidence they authors should not make any detailed conclusions on Cd2+ binding but just state that the changes in enzyme activity induced by Cd2+ cannot be reversed by washing. 

The conditions for Cyclic Voltammetry are not very clear.  For  is N2 purging necessary?  There is no information on cell volume and geometry of the cell. What was the scan-rate.  I assume the scan is stared at -0.2V and then is increased to 0.6V? I assume that the signal results from the oxidation and reduction of TBD. Is the presence of peroxide necessary to see the reversible oxidation and reduction of TBD?

The results presented indicate that the presence of the active enzyme inhibits the CV signal and the signal returns when Cd2+ is added.  When the authors plot the inhibition, what exactly are they plotting for degree of inhibition since the signal increases with Cd2+? The effect here is opposite to the colorimetric approach, since the signal increases with Cd2+ concentration.  This needs clarification.

An interesting question is why does the presence of HRP inhibit the voltammetry signal? Is the unaltered enzyme blocking access to the electrode surface?

In figure 3, was the membrane present in all of the scan? Even the scan with just TBD/H202?  Figure 4 A is very hard to read. 

There is not much detail regarding the water sample or samples that were examined.  What would be the impact of a chelator on this assay?  For example, if EDTA or humic acids were present, would that prevent Cd2+ from interacting with HRP?  The EDTA test would be very simple to perform, unless EDTA also impacts HRP.

Reviewer 4 Report

Colorimetric/electrochemical detection using smart surfaces is gaining attention in recent year. R. Attaallah et al. developed enzymic membrane for the detection of Cd2+ using horseradish peroxide. The technique shows high sensitivity in both colorimetric and electrochemical detection. Authors have previously developed paper device for the cadmium detection. The study includes all necessary techniques for the detection of cadmium ions. However, I have few suggestions and corrections to improve the quality of the paper. I recommend for the acceptance of the manuscript with minor revisions answered the below comments.

  1. I suggest authors to add a brief explanation on colorimetric detection, their merits, current trends in introduction section. Please cite these papers in the reference:
  • https://doi.org/10.1016/j.teac.2021.e00152
  • https://doi.org/10.1016/j.aca.2021.338439
  • https://doi.org/10.1016/j.jphotochem.2022.113817
  1. Page no. 3, Line no. 110, the word in vitro should be italic.
  2. Reviewer suggest proposing the schematic interaction mechanisms of Cd2+, followed by their reversibility in presence of Cu2+ for better understanding for the readers.
  3. Reviewer is curious to know the interference of anions in the detection process.

Round 2

Reviewer 1 Report

I am ok with the authors reply and the modifications made in the manuscript.

Reviewer 2 Report

The revised version can be accepted for publication.